# Formative independent evaluation of a digital change programme in the English National Health Service: study protocol for a longitudinal qualitative study

Kathrin Cresswell [iD],[1] Aziz Sheikh,[1] Bryony Dean Franklin,[2] Marta Krasuska,[1] Hung Nguyen,[3] Susan Hinder,[3] Wendy Lane,[4] Hajar Mozaffar,[5] Kathy Mason,[4] Sally Eason,[4] Henry Potts,[6] Robin Williams [iD] [3]

For numbered affiliations see end of article.

**Correspondence to**
Dr Kathrin Cresswell;
Kathrin.Cresswell@ed.ac.uk

## ABSTRACT

**Introduction**  Many countries are launching large-scale, digitally enabled change programmes as part of efforts to improve the quality, safety and efficiency of care. We have been commissioned to conduct an independent evaluation of a major national change programme, the Global Digital Exemplar (GDE) Programme, which aims to develop exemplary digital health solutions and encourage their wider adoption by creating a learning ecosystem across English National Health Service (NHS) provider organisations.

**Methods and analysis**  This theoretically informed, qualitative, longitudinal formative evaluation comprises five inter-related work packages. We will conduct a combination of 12 in-depth and 24 broader qualitative case studies in GDE sites exploring digital transformation, local learning and mechanisms of spread of knowledge within the Programme and across the wider NHS. Data will be collected through a combination of semistructured interviews with managers, implementation staff (clinical and non-clinical), vendors and policymakers, plus non-participant observations of meetings, site visits, workshops and documentary analysis of strategic local and national plans. Data will be analysed through inductive and deductive methods, beginning with in-depth case study sites and testing the findings against data from the wider sample and national stakeholders.

**Ethics and dissemination**  This work is commissioned as part of a national change programme and is therefore a service evaluation. We have ethical approval from the University of Edinburgh. Results will be disseminated at six monthly intervals to national policymakers, and made available via our publicly accessible website. We will also identify lessons for the management and evaluation of large-scale evolving digital health change programmes that are of international relevance.

## INTRODUCTION

Healthcare systems internationally strive for excellence. Excellence in health systems today is increasingly conceptualised in terms

### Strengths and limitations of this study

► A strength is that we will attempt to balance depth and breadth through conducting both detailed embedded case studies and 'lighter touch' studies in a broader sample of provider organisations.
► The formative nature of the work means that the research team is planning to play an active role in shaping implementation strategy and the ongoing implementation of the Global Digital Exemplar (GDE) Programme, presenting a significant strength in terms of relevance and verification for decision-makers.
► A limitation is that the qualitative nature of the study can provide only limited insights into outcomes emerging during the course of the Programme and further change over longer timeframes than the evaluation. It may also be difficult to disentangle the impact of the GDE Programme from other transformation initiatives running concomitantly.

achieving the 'triple aim' of better health outcomes, better value and better patient experience.[1] Policy initiatives throughout the economically developed world have sought to speed up the journey to achieve these challenging goals through various digitisation strategies. These include, for instance, the Health Information Technology for Economic and Clinical Health (HITECH) Act in the USA, the National Programme for Information Technology (NPfIT) in England, and Australia's National Digital Health Strategy and Framework for Action.[2–4]

However, these strategies have shown varying levels of success. For instance, although the HITECH reform was successful in getting organisations to adopt electronic health records (EHRs), this and other studies

have largely failed to demonstrate clinical benefits from these systems.[5] Similarly, the envisioned large-scale EHR adoption through centralised procurement of systems in the NPfIT in 2002 yielded unintended consequences, with early EHR systems showing difficulty fulfilling organisational and user needs, which ultimately led to a change in strategic direction in favour of more localised decision-making.[4] Digitisation without central direction between 2011 and 2016 was also not very successful in England, as individual healthcare organisations had limited resources and capacity to implement and optimise digital systems.[6] Projects had focused on deployment rather than wider service improvement and a lack of standardisation threatened the interoperability agenda.[7]

In 2016, the UK government therefore commissioned the US physician Robert Wachter to lead an independent review of the state and future strategic direction of digital health strategy in England.[8] One of the key recommendations from this was to selectively invest available resources to create a cohort of digital centres of excellence. Consequently, in 2017, National Health Service (NHS) England launched a flagship Global Digital Exemplar (GDE) Programme.[9] The GDE Programme is a £395 million national investment designed to establish selected digitally advanced provider organisations through funding and partnership opportunities to become Exemplars over 2–3.5 years.[10] These provider organisations in the GDE Programme cover a variety of care settings including acute care, specialist care, mental health and ambulance services. The underlying assumption was that digitally advanced sites would become international centres of excellence and create best practice models and learning for later implementers. GDE provider organisations (henceforth referred to as GDEs) were paired with somewhat less mature Fast Follower (FF) provider organisations to apply these advances. GDEs and FFs would capture best practice models and lessons in 'Blueprints', which would be disseminated within and beyond the Programme to accelerate the spread of this learning nationally. NHS England commissioned our team to evaluate this initiative.

The aim of our work, which has commenced in 2018 and is due to complete in 2021, is to conduct a formative evaluation of the GDE Programme. This includes exploring digital transformation in GDEs, the spread of learning among GDEs and FFs, and the establishment of a broader learning ecosystem. We will work jointly with NHS England and GDEs/FFs to discuss the implications of our findings and help the GDE Programme in achieving its vision. This will help to ensure that appropriate infrastructure and leadership is in place for sites to achieve international digital excellence.

## METHODS AND ANALYSIS

We will conduct a longitudinal qualitative formative evaluation, where GDEs and FFs will be conceptualised as case studies.[11] This format allows us to explore implementation, adoption and optimisation processes in context and to extract potentially transferable lessons associated with developments over time. For the purposes of evaluating the GDE Programme, we conceptualise each provider organisation as a case, where we can analyse context, processes and outcomes. We expect that each case will include a range of small-scale technology innovations as well as, in some instances, renewal of EHR infrastructures. We have significant experience with the case study method and have employed it successfully in previous work investigating large-scale digital health change programmes.[4 12]

Our work will take place in five complementary work packages (WPs), summarised in figure 1.

### Patient and public involvement
No patient was involved.

### Setting and participants
There are a total of 23 GDEs and 24 FFs in the Programme. We will collect in-depth data from a subset of 12 sites, and high-level data from the remainder. The in-depth sites will be sampled purposefully for maximum variation to represent a range of settings (eg, acute, mental health, specialist), core EHR infrastructures, geographical locations, sizes, implementation timelines and levels of digital maturity. In doing so, we will seek representation of sites with large commercial integrated and best-of-breed systems; sites located in the South, Midlands and North of England; teaching and non-teaching provider organisations; and comparatively low, medium and high levels of baseline digital maturity. GDEs and FFs will be included. Individual participants will include programme management staff within provider organisations (clinical and non-clinical), system vendors and national stakeholders (eg, programme managers and policymakers).

### Overall study design
We will undertake in-depth qualitative investigations in 12 provider organisations purposefully selected from all acute, specialist and mental health GDEs and FFs (WP2 in figure 1). Ambulance organisations will be excluded as these were out of scope for this commission. We will complement these in-depth sites with more selective data collection across the entire sample of GDEs and FFs (WP1 in figure 1), in order to balance depth of findings with the breadth of insights required to draw meaningful conclusions. Work in study sites will be complemented by data collection from the wider healthcare community, policymakers, vendors and the international community (WP5 in figure 1).

We will use qualitative methods (comprising semistructured interviews, observations and documentary analysis) to gather data on technology selection, implementation and adoption, change management strategy, governance processes and stakeholder engagement. We will also seek to explore the impact of contextual factors on change processes to facilitate the identification of critical success factors and

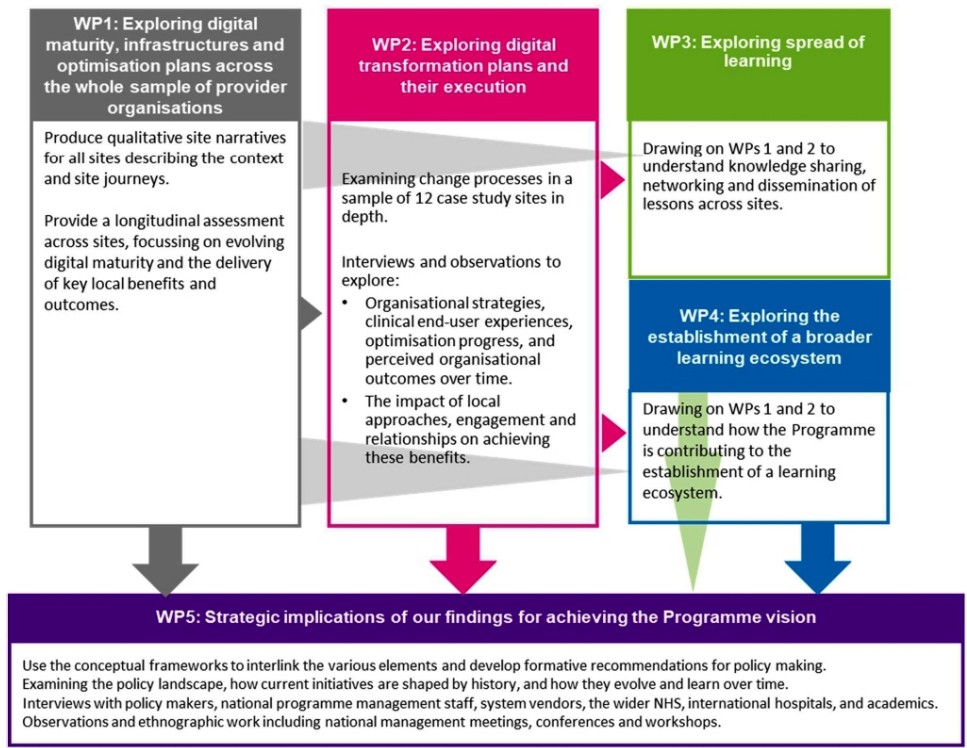

**Figure 1** High-level overview of our methods in each of the five work packages (WPs). NHS, National Health Service.

dependencies so that we can provide outputs that have practical application to accelerate uptake and impact locally and nationally.

### Analytical framework

A conceptual/analytical framework and methodology informed by pertinent contemporary theoretical developments is important to guide the research and generate generalisable insights for policy and practice. We will therefore draw on a pragmatic application of a number of theories (box 1).[13–15] This approach has been successfully applied in our previous work and enables us to build on existing knowledge through obtaining theoretical insights (and thereby allowing generalisations) without neglecting the more immediate need to provide formative strategic input.[4] In integrating these approaches, we will explore how various technological systems and social structures coevolve over time shaping each other throughout a continuous process. This will be achieved

through applying a theory-informed coding framework developed in related work (see the Data analysis section).

Our formative evaluation will provide insights into how the continuing development of the GDE Programme may be enhanced to promote positive impacts on provider digitisation and patient outcomes. We will work closely with policymakers to develop a detailed understanding of the existing stakeholders, policy landscape and evolving approaches to Programme management, so that we can avoid duplicating the significant efforts made to monitor substantive outcomes. This detailed understanding of processes will help us to refine our overall approach, focusing on emerging local and national priorities while being mindful of implementation timelines.

We now describe the methods used in each of the WPs in more detail.

### WP1: exploring digital maturity, infrastructures and optimisation plans across all provider organisations taking part in the GDE Programme

#### Objectives

GDEs and FFs are at various stages of system implementation and optimisation, with a range of different information infrastructures in place. In this WP, we seek to make assessments surrounding the success of the GDE Programme and gain insights into progress (or lack of).

#### Design

In this WP, we will collect qualitative descriptive data from the acute and mental health GDEs and FFs that are not selected for WP2 in-depth case studies.

---

**Box 1 Conceptual approaches that we propose to draw on**

Sociotechnical considerations—paying attention to social, technological and organisational processes and exploring how these influence each other over time.
An evolutionary perspective encompassing the evolving technology lifecycle—technology implementation, adoption and optimisation unfolds gradually over time offering opportunities for learning. These need to be examined over extended timeframes.[26]
Information infrastructures—how technologies emerge and how they together form 'systems of systems'.[27]

---

## Sampling

We will include all acute and mental health GDEs and FFs in this WP and purposefully sample members of the local programme team who have insights into existing systems and strategies (including chief information officers and their GDE management teams). Sites will be approached through our existing contacts at Arden and GEM Commissioning Support Unit, who are part of our team and already have established gatekeeper contacts.

## Data collection

Data collection will consist of gathering and analysing a range of documentation including funding agreements detailing provider organisations' transformation plans, strategies and digital maturity assessments and conducting a series of one-to-one in-depth semistructured face-to-face or telephone interviews, group interviews (where preferred by sites) and site visits (see box 2 for indicative topic guides). We will produce summaries describing the organisational context, technological systems and digital strategies in each site. In order to assess individual journeys over time and to capture a longitudinal dimension, we will visit sites at the start of their GDE Programme and revisit sites at least 6 months after the implementation of GDE-related systems to gain insights into the evolving digital maturity and the delivery of key local benefits and outcomes.

## WP2: exploring digital transformation plans and their execution

### Objectives

To measure progress in a more focused way, we will examine change processes and specific clinical outcomes in selected settings in depth.

### Design

We will use a combination of qualitative interviews and non-participant observation of strategic meetings to explore organisational strategies, clinical end user experiences, implementation/use/optimisation progress, and perceived individual/organisational benefits/outcomes over time (box 2). We aim to investigate perceived outcomes, so it is difficult to anticipate what these may be in advance. We expect that many will be qualitative as quantitative outcomes tend to materialise over long timeframes. Patient outcomes, in particular, are unlikely to emerge during the conduct of this work but we may observe some improvement in organisational performance.[16]

### Sampling

This WP is concerned with insights into change processes in a sample of 12 purposefully selected case study sites, aiming for maximum variation as outlined above.

Within each site, we will sample participants purposefully to represent a range of viewpoints (eg, different clinical and managerial backgrounds) and levels of seniority. Gatekeepers will be approached to help us establish initial contacts and we will snowball sample based on these.

---

**Box 2  High-level interview guide**

**Background**
► Background and role of interviewee(s) (work package 1 (WP1), WP2).
► Digital trajectory/journey before the Programme (WP1, WP2).

**Strategy**
► Details of change/implementation strategy and benefits realisation strategy (WP1, WP2).
► Implementation approach (resources, leadership, engagement, training, sustainability) (WP1, WP2).

**Implementation progress**
► Details of new digital functions being introduced as part of the Programme and other current/recent changes (WP1, WP2).
► Progress in implementing these (WP1, WP2).
► Issues arising in implementation (WP1, WP2).
Overall thoughts on the Programme (rationale, aims, how it has gone so far and what could be done better) (WP2).

**Benefits realisation and reporting (WP2)**
► Benefits achieved through functionalities.
► Challenges in realising these benefits.
► Facilitators for achieving benefits.

**Blueprinting**
► Overview of Blueprint production and use (WP1, WP2).
► Experiences of the Blueprinting process (challenges, areas for improvement) (WP2).

**Knowledge management, networking and learning (formal and informal)**
► Existing networks/learning and key stakeholders (within the Programme and outside the Programme) (WP1, WP2).
► Relationship between Fast Follower (FF) and Global Digital Exemplar (GDE) organisations (WP1, WP2).
► Experiences and perceptions on what knowledge networks are most useful and why (WP2).
► Other relationships/sources of information (WP2).
► Perception of how national support can promote knowledge exchange and networking (WP2).

**Vendors (WP2)**
► Relationship with vendors.
► Views on national digital health infrastructures.

**Lessons learnt and way forward**
► Key lessons learnt to date (WP1, WP2).
► Perceptions around what support is needed (WP2).
► Best ways to spread learning (WP2).
► View on the sustainability of benefits (WP2).
► Perception of if/how benefits have been realised (WP2).
► Unintended consequences (WP2).

---

As participants will need to have insights into the GDE Programme, we expect to focus sampling on members of local strategic committees and information technology management staff. We will stop recruiting new participants when no new themes are emerging and when we have reached thematic saturation.[17]

### Data collection

Data collection will consist of a combination of one-to-one semistructured face-to-face or telephone interviews, group interviews (where preferred by participants), observations of GDE-related meetings and workshops, and collection of documents. Designated lead researchers

will collect data in in-depth case study sites in order to allow immersion in the setting.

Researchers will audio record interviews and group interviews and prepare accompanying field notes. A professional transcribing service will produce transcripts of these recordings. Interviews will allow us to gain detailed insights into participant attitudes towards the Programme, expectations, local complexities, perceived benefits, unexpected consequences, challenges experienced, and lessons learnt.

Lead researchers will conduct non-participant observations either in person or online. This approach will allow us to understand dynamics within sites (eg, when observing meetings of local management groups). During observations, researchers will take detailed field notes relating to content, social dynamics and their own impressions by considering the observation within the wider context of the overall evaluation work.

In addition to interviews and observations, we will also collect local documents that will allow us to understand strategies and implementation/optimisation plans. We will use these as contextual background reading to inform interview topic guides and interpretations of observations.

## WP3: exploring spread of learning
### Objectives
To explore knowledge transfer and dissemination of lessons and networking activity across GDE and FF sites.

### Design
We will undertake secondary analysis of data collected in WP1 and WP2 to explore mechanisms associated with knowledge transfer. This will draw on qualitative data collected in WP1 and WP2 to extract spread and sharing of knowledge between sites through formal and informal mechanisms produced through targeted programme activities identified in the analysis of documents. Key lines of enquiry will include exploring instances where knowledge transfer and spread was perceived as successful/unsuccessful and exploring the underlying reasons why.

## WP4: exploring the establishment of a broader learning ecosystem
### Objectives
Here, we seek to understand how the Programme is contributing to the establishment of a wider digital health learning ecosystem within and beyond the GDE Programme, including both the formal knowledge transfer mechanisms planned under the Programme and informal knowledge exchanges that may emerge. We conceptualise a learning ecosystem as interorganisational knowledge transfer and learning that occurs over time across the entire health system (ie, not only the GDE sites).

### Design
We will undertake a secondary analysis of formal and informal means of sharing knowledge identified in WP3,

and of data collected in WP1 and WP2 to examine the formation and operation of learning and knowledge networks across the GDE Programme and with the wider NHS and other communities. Key lines of enquiry will include examining stakeholder experiences and overall patterns to address the (variable) dynamism of learning, and the incentives for and barriers to effective knowledge transfer.

## WP5: strategic implications of our findings for achieving the Programme vision
### Objectives
This final WP is concerned with the integration and dissemination of findings from the evaluation. We will work to connect the results from WP1 to WP4, with a view to mapping out the wider overall picture and establishing the enduring themes that offer useful insights to those who will plan, manage and participate in future digital health deployments.

### Design
This WP will be a qualitative longitudinal study comprising qualitative interviews, observations and collection of documents. Discussions with key stakeholders will examine how historical and contextual factors shape the processes underway and help explicate implications of emerging findings for policy.

### Sampling
In this final WP, we will engage with a wide range of stakeholders including policymakers, national programme management staff, system vendors, the wider NHS, international hospitals and partner organisations, and academics. These will be recruited with the help of key national gatekeepers in our Steering Group or approached directly by us via publicly available email addresses.

### Data collection
We will conduct one-to-one semistructured interviews with researchers taking detailed field notes. In addition, we will also conduct ethnographic fieldwork including attending all national programme management meetings, and national conferences and workshops that are related to the GDE Programme. Collection of national strategic plans will complement interviews and observations. This WP will help us position our findings within the existing policy landscape and within the history of digital change in the NHS. It will also allow exploring evolving strategies and changes over time. We will use our conceptual frameworks to interlink the various elements and develop formative recommendations for policymaking. These recommendations will be fed back through written reports and face-to-face meetings with senior policymakers, with whom we have established relationships.

### Data analysis
Data analysis will be iterative and feed into subsequent data collection, using a combination of deductive and

> **Box 3  Overview of categories in the Technology, People, Organizations and Macro-environmental factors (TPOM) evaluation framework[19]**
>
> **Technological factors**: usability; performance; adaptability and flexibility; dependability, data availability, integrity and confidentiality; data accuracy; sustainability; security.
>
> **Social/human factors**: user satisfaction; complete/correct use; attitudes and expectations; engagement; experiences; workload/benefits; work processes; user input in design.
>
> **Organisational context**: leadership and management; communication; timelines; vision; training and support; champions; resources; monitoring and optimisation.
>
> **Wider macroenvironment**: media; professional groups; political context; economic considerations and incentives; legal and regulatory aspects; vendors; measuring impact.

inductive methods.[18] We will develop a theory-informed coding framework in which lead researchers will code qualitative data from all WPs, while allowing additional categories to emerge. We will draw on the Technology, People, Organizations and Macro-environmental factors (TPOM) evaluation framework we have developed in related work (box 3). This includes various subcategories that will be used as prospective criteria against which assessments will be made.[19]

Documentary, observation and interview data will be collated for each case by the lead researcher and coded against the TPOM framework, allowing additional categories to emerge. Documents, observations and interviews from WP5 will be analysed separately and integrated with findings from case studies. We will seek to feed back and test emerging findings into concurrent data collection.

We will use NVivo software V.11 to facilitate the process of coding qualitative data.[20]

During three monthly intensive analysis meetings with the wider team (ie, all of the authors), we will discuss emerging findings and distil implications for policy-making. These will then be collated and synthesised for feedback to the Steering Group of the evaluation, which comprises senior national programme managers and internationally renowned academics. The role of this group will be to consider this formative feedback regularly and (where relevant) incorporate insights into strategic decision-making. Members will also help to direct the research towards areas where it can achieve maximum impact.

Analysis meetings will initially have a relatively broad focus, with increasing depth over time, focusing in on issues identified as important by the Steering Group and the research team. In line with the aims of this work, we will initially explore digital transformation within sites, before analysing spread of learning across GDE and FF sites, and then analyse how the Programme has helped (or not) to establish a wider learning digital health ecosystem. We will focus on challenges and unanticipated consequences in most detail. The in-depth case studies will allow us to get detailed insight into local dynamics that we will then test across the wider sample, seeking confirming and disconfirming evidence.

## ETHICS AND DISSEMINATION

This work is a service evaluation of a national programme and therefore does not require review by an NHS research ethics committee. We received institutional ethical approval from the School of Social and Political Science Research Ethics Committee at The University of Edinburgh, UK. We will adhere to good practice and relevant ethical guidelines in obtaining verbal informed consent for participation, as well as anonymising individuals and sites prior to any dissemination. Data will be stored on university servers.

We will submit written reports of our emerging findings to our quarterly Steering Group meetings. We will also seek to publish the written reports on our publicly accessible website.[21] In addition, we will develop a detailed publication strategy for validating and disseminating key findings in academic peer-reviewed journals.

## STRENGTHS AND LIMITATIONS

Conducting a combination of broad and in-depth case studies will allow us to balance breadth and depth. A further strength is the formative nature of this work, where the research team will seek to play an active role in shaping the strategy and ongoing implementation of the GDE Programme. However, a limitation is that the qualitative methods used for formative evaluation are unlikely to provide detailed substantive information about the impact/eventual outcomes of the programme (which may be difficult to disentangle from the impact of other initiatives). We may also encounter difficulties as the GDE Programme is still unfolding and may be subject to delays and/or changes in direction. This may require flexibility in the implementation of the evaluation design.

## CONTRIBUTIONS TO THE LITERATURE

Although digital health change programmes are increasingly large scale, there is a dearth of evidence around how these often evolving programmes can be managed and evaluated in order to maximise their benefits.[22] The initiative being studied represents the largest attempt to create a concerted digital learning ecosystem. There may be a missed opportunity to learn from previous large-scale initiatives both nationally and internationally.[23] For example, the English NHS has been involved in a range of initiatives to promote digitisation with varying levels of success over the last 20 years but key tensions, for example, around balancing national and local ownership and priorities, have not yet been resolved.[4] This work will, we hope, help to address this gap and also allow to

identify factors which may impede or accelerate digitisation, characteristics of learning, knowledge flows and associated networks.

Our evaluation will also contribute to discussions around conceptualising digital maturity, a concept that has, to date, been poorly defined but is needed by policymakers and programme managers to establish baselines and demonstrate progress of change initiatives.[24] We hope to advance the literature in defining the concept, highlight emerging issues, and develop implications for measuring digital maturity in hospitals.

In-depth case studies will further help to shed light on ongoing tensions in the process of digital transformation and associated contexts, mechanisms and outcomes.[25] Of particular interest will be areas where there are no identified 'recipes for success' such as leadership, clinical engagement, vendor market management, and governance.[15]

The evaluation will also identify internationally relevant lessons that may inform attempts to establish digital health learning ecosystems elsewhere. Organisational learning in health systems and knowledge flows have received little attention within the healthcare domain to date, but this area is likely to gain importance as concerted efforts to develop learning ecosystems will increase internationally in order to promote learning and accelerate digitally enabled change.

## CONCLUSIONS

The GDE Programme is the first concerted effort to develop a national digital health-learning ecosystem. Our real-time national evaluation of this initiative provides an important opportunity to feed research findings into policymaking and thereby maximise impact.

**Author affiliations**
[1]Usher Institute, University of Edinburgh, Edinburgh, UK
[2]School of Pharmacy, University College London, London, UK
[3]School of Social and Political Science, University of Edinburgh, Edinburgh, UK
[4]NHS Arden and Greater East Midlands Commissioning Support Unit, Warwick, UK
[5]Business School, University of Edinburgh, Edinburgh, UK
[6]Centre for Health Informatics and Multiprofessional Education, University College London, London, UK

**Acknowledgements** We gratefully acknowledge the input of the Steering Group of this evaluation.

**Contributors** KC, RW and AS conceived this paper. KC and RW led the drafting of the manuscript. BDF, MK, HN, SH, WL, HM, KM, SE and HP commented on drafts of the manuscript.

**Funding** This article has drawn on a programme of independent research funded by NHS England. This work was also supported by the National Institute for Health Research (NIHR) Imperial Patient Safety Translational Research Centre.

**Disclaimer** The views expressed in this publication are those of the authors and not necessarily those of the NHS, NHS England, NHS Digital, the NIHR or the Department of Health and Care.

**Competing interests** All authors are investigators on the evaluation of the GDE Programme (https://www.ed.ac.uk/usher/digital-exemplars). AS was a member of the Working Group that produced *Making IT Work*, and was an assessor in selecting GDE sites. BDF supervises a PhD student partly funded by Cerner, unrelated to this paper.

**Patient and public involvement** Patients and/or the public were not involved in the design, or conduct, or reporting, or dissemination plans of this research.

**Patient consent for publication** Not required.

**Provenance and peer review** Not commissioned; externally peer reviewed.

**ORCID iDs**
Kathrin Cresswell http://orcid.org/0000-0001-6634-9537
Robin Williams http://orcid.org/0000-0002-9044-4611

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
