## [Reviewer comments · BMJ Open]

ARTICLE DETAILS

TITLE (PROVISIONAL)	The formative independent evaluation of a digital change programme in the English National Health Service: Study protocol for a longitudinal qualitative study
AUTHORS	Cresswell, Kathrin; Sheikh, Aziz; Dean Franklin, Bryony; Krasuska, Marta; Nguyen, Hung; Hinder, Susan; Lane, Wendy; Mozaffar, Hajar; Mason, Kathy; Eason, Sally; Potts, Henry; Williams, Robin

VERSION 1 – REVIEW

REVIEWER	Simon Mathews Johns Hopkins University, Baltimore, USA
REVIEW RETURNED	05-Jul-2020

GENERAL COMMENTS	The authors taken on the ambitious task of providing a framework for evaluating the Global Digital Exemplar (GDE) Programme. This is important work and should be completed. After reviewing the manuscript I have the general impression that an extensive qualitative assessment will be undertaken across a number of relevant topic areas, across a range of stakeholders and settings. This data will then be abstracted to provide a descriptive and thematic understanding of how these GDEs and FFs function, which will inform policy and facilitate learning and dissemination of best practices. This will undoubtedly be useful, however, from an evaluation perspective it would also be helpful to identify criteria (prospectively) that will be used as part of the assessment. For example, can a checklist or scoring rubric be created to ensure standardized assessment across locations and settings? Please consider incorporating assessing technology specific issues such as: interoperability; usability; privacy; impact on workflow and culture. In addition, can you provide additional detail on the following: 1. Criteria for selecting the 12 in depth evaluations. Is this just based on solely on the settings described or additional rationale? Will they be chosen randomly?2. An explanation of how the conceptual approaches outlined in Box 1 are integrated/operationalized in the evaluation?3. Why case studies format was chosen as the preferred approach. What will the case study entail? Is there a similar format that can be referenced?4. What clinical outcomes will be evaluated in WP-2 and how will they be selected and measured?5. Can WP-4 be embedded into WP-3 as they are related and there is not much detail provided for the former?6. What is the process for how the findings identified in WP-5 will inform policy? Are there key stakeholders or forums where assessment will be presented?
---

REVIEWER	Pirkko Nykänen Tampere university, Finland
REVIEW RETURNED	27-Jul-2020

GENERAL COMMENTS	The study is generally well planned, strengths and weaknesses are identified adequately. This a plan for a qualitative study to evaluate digital transformation, spread of learning and establishment of a learning ecosystem, and it is a difficult task to perform this kind of qualitative study when 'ceteris paribus' assumption does not hold - this has been identified in the protocol. Some aspects in the study protocol need clarification, e.g. Wp2: detailed selection criteria for 12 sites to be studied in depth and broad, Wp3 and Wp4: qualitative data analysis methods. This is a major weakness in the protocol. Data is planned to be collected with many different methods and data analysis is a crucial step - more details on the analysis methods are needed. It also remains unclear how the ecosystem establishment will be perceived in Wp4, what are interesting features of the ecosystem. The protocol does not explicate when the study is planned to start and end, and in introduction there could be short summary of the status of GDEs and FFs before the study start.
------------------	--

VERSION 1 – AUTHOR RESPONSE

Reviewer 1

Point 1: The authors taken on the ambitious task of providing a framework for evaluating the Global Digital Exemplar (GDE) Programme. This is important work and should be completed. After reviewing the manuscript I have the general impression that an extensive qualitative assessment will be undertaken across a number of relevant topic areas, across a range of stakeholders and settings. This data will then be abstracted to provide a descriptive and thematic understanding of how these GDEs and FFs function, which will inform policy and facilitate learning and dissemination of best practices. This will undoubtedly be useful, however, from an evaluation perspective it would also be helpful to identify criteria (prospectively) that will be used as part of the assessment. For example, can a checklist or scoring rubric be created to ensure standardized assessment across locations and settings? Please consider incorporating assessing technology specific issues such as: interoperability; usability; privacy; impact on workflow and culture.

Response: When analysing the data, we will draw on the Technology, People, Organisations, and Macro-environmental factors (TPOM) evaluation framework we have developed in related work. This has recently been published and includes various sub-categories that will be used as prospective criteria against which assessments will be made.^[1] We now describe this in more detail in the Data Analysis section and provide a summary of categories and sub-categories in Box 3.

Point 2: Criteria for selecting the 12 in depth evaluations. Is this just based on solely on the settings described or additional rationale? Will they be chosen randomly?

Response: As described, we will select sites for maximum variation to reflect a range of settings (e.g. acute, mental health, specialist) core electronic health record infrastructures, geographical locations, sizes, implementation timelines, and levels of digital maturity. In doing so, we will seek representation of sites with large commercial integrated and Best-of-Breed systems; sites located in the South, Midlands and North of England; teaching and non-teaching provider organizations; and comparatively low, medium- and high-levels of baseline digital maturity. We have now described these criteria in more detail and made clearer that we will select sites purposefully.

Point 3: An explanation of how the conceptual approaches outlined in Box 1 are integrated/operationalized in the evaluation?

Response: We have revised the Analytical Framework section describing how we, in integrating these approaches, will explore how various technological systems and social structures co-evolve over time shaping each other throughout a continuous process. We also now cross-reference the TPOM evaluation framework described in the analysis section, as this will help us to integrate these theoretical perspectives.

Point 4: Why case studies format was chosen as the preferred approach. What will the case study entail? Is there a similar format that can be referenced?

Response: We now explain that the case study format has been chosen, as it allows to explore implementation, adoption and optimisation processes in context. This format allows us to explore implementation, adoption and optimisation processes in context and to extract potentially transferable lessons associated with developments over time.^[2] For the purposes of evaluating the GDE Programme, we conceptualise each provider organisation as a case, where we can analyse context, processes and outcomes. We expect that each case will include a range of small-scale technology innovations as well as, in some instances, renewal of electronic health record infrastructures. We have significant experience with the case study method and have employed it successfully in previous work investigating large-scale digital health change programmes.^[3]

Point 5: What clinical outcomes will be evaluated in WP-2 and how will they be selected and measured?

Response: We aim to investigate perceived outcomes, so it is difficult to anticipate what these may be in advance. We expect that many will be qualitative as quantitative outcomes tend to materialize over long timeframes.^[4] Patient outcomes in particular are unlikely to emerge during the conduct of this work but we may observe some improvement in organisational performance. We now elaborate on this in the Design section of WP2.

Point 6: Can WP-4 be embedded into WP-3 as they are related and there is not much detail provided for the former?

Response: We would prefer to leave WP3 and WP4 as they are, although related, conceptually distinct. WP3 is exploring inter-organisational knowledge transfer, and WP4 is designed to explore the emergence of a learning ecosystem (which may for example relate to broader developments beyond the GDE sites, such as supplier user-groups). This feature is a distinctive feature of the GDE Programme.

Point 7: What is the process for how the findings identified in WP-5 will inform policy? Are there key stakeholders or forums where assessment will be presented?

Response: We have added to the Data Collection section of WP5 that our recommendations will be fed back through written reports and face-to-face meetings with senior policy makers, with whom we have established relationships.

Reviewer 2

Point 1: Some aspects in the study protocol need clarification, e.g. Wp2: detailed selection criteria for 12 sites to be studied in depth and broad.

Response: This point is related to Point 2 raised by Reviewer 1. We now provide detailed information on how we will chose case study sites in the Settings and Participants section and also cross-reference to this section in WP2.

Point 2: Wp3 and Wp4: qualitative data analysis methods. This is a major weakness in the protocol. Data is planned to be collected with many different methods and data analysis is a crucial step - more details on the analysis methods are needed.

Response: This point is related to Point 1 raised by Reviewer 1. We now include a detailed description of our theory-informed coding framework in the Analysis section, and also link this to the Analytical Framework section. In addition, we now provide more detail on how we plan to consolidate analysis of data collected across data collection methods. Documentary, observation, and interview data will collated for each case by the lead researcher and coded against the TPOM framework, allowing additional categories to emerge. Documents, observations, and interviews from WP5 will be analysed separately and integrated with findings from case studies. We will seek to feed back and test emerging findings into concurrent data collection.

Point 3: It also remains unclear how the ecosystem establishment will be perceived in Wp4, what are interesting features of the ecosystem.

Response: We now clarify in WP4 that we conceptualise a learning ecosystem as inter-organisational knowledge transfer and learning that occurs over time across the entire health system (i.e. not only the GDE sites).

Point 4: The protocol does not explicate when the study is planned to start and end.

Response: We now clarify in the Introduction section that our work has commenced in 2018 and is due to complete in 2021.

Point 5: In the introduction there could be short summary of the status of GDEs and FFs before the study start.

Response: We have amended the Introduction section to clarify that the GDE Programme was designed to establish selected digitally advanced provider organisations (i.e. these organisational labels were a feature introduced by the Programme).

We have also revised the format of Figure 1, as requested by the editor.

VERSION 2 – REVIEW

REVIEWER	Pirkko Nykänen Tampere university, Faculty for Information Technology and Communication Sciences
REVIEW RETURNED	13-Aug-2020

GENERAL COMMENTS	The paper has been revised on the basis of the reviewers' comments. The current version has much improved. My earlier worry on the selection criteria of 12 GDEs for inclusion into this study has now been sufficiently addressed. The problem with data analysis methods has been well addressed now with the introduction of the TPOM framework with sufficient details. Also the duration and starting of the study has been presented. With the revision the paper now presents a detailed and justified description of the study. As it is now I recommend the paper ready for publication now. The scientific community looks forward to see the results, conclusions and recommendations from this study.
---